# OpenReview forum: "Variational Quantum Circuits for Efficient Transformer Attention"
_purdue.edu/Purdue_University/PQAI/2025/Symposium — PQAI 2025 Poster_

### Official Review · Reviewer_zeSg · 2025-07-25
**A well written paper with a few weaknesses, well suited to a poster session**

**Rating:** 5
**Confidence:** 5

**Review:**

This paper presents a quantum implementation of the attention function that forms the basis of transformer models. The paper is well written, despite not clearly citing prior work -- a few citations are included at the end but are not referenced throughout the work -- it provides an overview of prior work describing both classical transformer architectures and touching on recent progress in quantum machine learning. The quantum-enhanced transformer is implemented by a combination of quantum and classical machine learning models, the classical component consists of feed-forward layers although it is not exactly clear where these are used in relation to the quantum model. The quantum model consists of two embedding circuits which embed the query and key vectors, followed by a CNOT ladder to entangle the query and key registers. This, in turn, is followed by a variational circuit using a hardware-efficient ansatz. The model is trained using the parameter-shift rule to calculate gradients, with simulations conducted via Pennylane+Pytorch on the 20newsgroups dataset.

At a high level, the paper tackles an interesting problem --- how can quantum computers improve upon the attention-based ML architectures that have shown so much promise in recent years? --- but it is lacking with respect to a few key details.

1. The paper does not adequately cite/acknowledge prior work, particularly in the domain of quantum natural language processing where frameworks like DisCoCat (https://arxiv.org/abs/1608.01406) have been explored by prior work related to quantum NLP.
2. Although the text claims to present a quantum-enhanced transformer architecture that can be easily integrated into existing architectures without large-scale redesigns, the approach suffers from a number of drawbacks that are not addressed in sufficient detail to form a convincing argument:
    - The approach relies on an amplitude embedding of classical data vectors, which in the general case, require exponentially deep circuits to prepare the state vector.
    - The approach relies on variational circuits which are known to suffer from trainability issues (a topic not addressed by the current paper). Indeed, the parameter shift rule actually increases the overhead of training with respect to classical models since it requires a linear number of circuit evaluations (proportional to the total number of parameters in the model) in order to update the entire gradient. These issues are especially prevalent for ansatzes, such as the hardware-efficient ansatz, that may be very expressive (the paper repeatedly mentions the exponentially large feature space accessible by quantum computers) but often leads to poor training due to barren plateaus and local minima.
    - Finally, the evaluation presented in Section 6 is not sufficient to back up the claims of (potential) quantum advantage. How many parameters are used in the classical models in Table 1? This would help contextualize the benchmark comparison. It is also difficult to see significant differences in performance since no error bars or statistical analysis is presented for the benchmark, which may only consist of a single training run. The discussion on the entanglement entropy measured in the output state of the quantum circuits is interesting, but does not go far enough to definitively show meaningful contributions to the attention computation.


Overall, the paper was well written and has the start of many good ideas. However, the evaluation in its current state is insufficient to draw concrete conclusions about the proposed architecture. Therefore I think this work may be best presented at a poster session.

---

### Official Review · Reviewer_vZ3B · 2025-07-26
**poster or oral recommanded**

**Rating:** 4
**Confidence:** 5

**Review:**

Thank you for submitting your work. The paper aligns well with the scope of the conference.

The paper shows original idea in using the quantum circuit evaluations for transformer models showing a novel quantum attention mechanism.

I appreciate the detail presented in the technical approach specifically in quantum and ML concepts. However the 12 qubit simulations and datasets are limitations and severe constraints

However the paper does not have any in-text citations which is problematic from a formatting perspective. Therefore I recommend this to be a poster or oral presentation.

The paper proposed future work in exploring NLP-specific quantum circuits, which needs more work explaining how to integrate quantum components into a LLM, and how benchmarking will work specifically.

---

### Decision · Program_Chairs · 2025-07-29

Accept (Poster)